# Neural Pruning via Growing Regularization

**Huan Wang, Can Qin, Yulun Zhang**,[*] **Yun Fu**
Northeastern University, Boston, MA, USA
{wang.huan, qin.ca}@northeastern.edu,
yulun100@gmail.com, yunfu@ece.neu.edu

## Abstract

Regularization has long been utilized to learn sparsity in deep neural network pruning. However, its role is mainly explored in the small penalty strength regime. In this work, we extend its application to a new scenario where the regularization grows large gradually to tackle two central problems of pruning: pruning schedule and weight importance scoring. (1) The former topic is newly brought up in this work, which we find critical to the pruning performance while receives little research attention. Specifically, we propose an $L_2$ regularization variant with rising penalty factors and show it can bring significant accuracy gains compared with its one-shot counterpart, even when the *same* weights are removed. (2) The growing penalty scheme also brings us an approach to exploit the Hessian information for more accurate pruning *without knowing their specific values*, thus not bothered by the common Hessian approximation problems. Empirically, the proposed algorithms are easy to implement and scalable to large datasets and networks in *both structured and unstructured pruning*. Their effectiveness is demonstrated with modern deep neural networks on the CIFAR and ImageNet datasets, achieving competitive results compared to many state-of-the-art algorithms. Our code and trained models are publicly available at https://github.com/mingsuntse/regularization-pruning.

## 1 Introduction

As deep neural networks advance in recent years LeCun et al. (2015); Schmidhuber (2015), their remarkable effectiveness comes at a cost of rising storage, memory footprint, computing resources and energy consumption Cheng et al. (2017); Deng et al. (2020). Neural network pruning Han et al. (2015; 2016); Li et al. (2017); Wen et al. (2016); He et al. (2017); Gale et al. (2019) is deemed as a promising force to alleviate this problem. Since its early debut Mozer & Smolensky (1989); Reed (1993), the central problem of neural network pruning has been (arguably) how to choose weights to discard, i.e., the weight importance scoring problem LeCun et al. (1990); Hassibi & Stork (1993); Molchanov et al. (2017b; 2019); Wang et al. (2019a); He et al. (2020).

The approaches to the scoring problem generally fall into two groups: importance-based and regularization-based Reed (1993). The former focuses on directly proposing certain theoretically sound importance criterion so that we can prune the unimportant weights once for all. Thus, the pruning process is typically one-shot. In contrast, regularization-based approaches typically select unimportant weights through training with a penalty term Han et al. (2015); Wen et al. (2016); Liu et al. (2017). However, the penalty strength is usually maintained in a small regime to avoid damaging the model expressivity. Whereas, a large penalty strength can be helpful, specifically in two aspects. (1) A large penalty can push unimportant weights rather close to zero, then the pruning later barely hurts the performance even if the simple weight magnitude is adopted as criterion. (2) It is well-known that different weights of a neural network lie on the regions with different local quadratic structures, i.e., Hessian information. Many methods try to tap into this to build a more accurate scoring LeCun et al. (1990); Hassibi & Stork (1993); Wang et al. (2019a); Singh & Alistarh (2020). However, for deep networks, it is especially hard to estimate Hessian. Sometimes, even the computing itself can be intractable without resorting to proper approximation Wang et al. (2019a). On this problem, we ask: *Is it possible to exploit the Hessian information without knowing*

---
[*]Corresponding author.

*their specific values*? This is the second scenario where a growing regularization can help. We will show under a growing regularization, the weight magnitude will naturally separate because of their different underlying local quadratic structure, therein we can pick the unimportant weights more faithfully even using the simple magnitude-based criterion. Corresponding to these two aspects, we will present two algorithms based on a *growing $L_2$ regularization* paradigm, in which the first highlights a better pruning schedule[1] and the second explores a better pruning criterion.

**Our contributions**. (1) We propose a simple yet effective growing regularization scheme, which can help transfer the model expressivity to the remaining part during pruning. The encouraging performance inspires us that the pruning schedule may be as critical as the weight importance criterion and deserve more research attention. (2) We further adopt growing regularization to exploit Hessian implicitly, without knowing their specific values. The method can help choose the unimportant weights more faithfully with a theoretically sound basis. In this regard, our paper is the first to show the connection between magnitude-based pruning and Hessian-based pruning, pointing out that the latter can be turned into the first one through our proposed growing regularization scheme. (3) The proposed two algorithms are easy to implement and scalable to large-scale datasets and networks. We show their effectiveness compared with many state-of-the-arts. Especially, the methods can work seamlessly for both filter pruning and unstructured pruning.

## 2 RELATED WORK

**Regularization-based pruning**. The first group of relevant works is those applying regularization to learn sparsity. The most famous probably is to use $L_0$ or $L_1$ regularization Louizos et al. (2018); Liu et al. (2017); Ye et al. (2018) due to their sparsity-inducing nature. In addition, the common $L_2$ regularization is also explored for approximated sparsity Han et al. (2015; 2016). The early papers focus more on unstructured pruning, which is beneficial to model compression yet not to acceleration. For structured pruning in favor of acceleration, Group-wise Brain Damage Lebedev & Lempitsky (2016) and SSL Wen et al. (2016) propose to use Group LASSO Yuan & Lin (2006) to learn regular sparsity, where the penalty strength is still kept in small scale because the penalty is uniformly applied to all the weights. To resolve this, Ding et al. (2018) and Wang et al. (2019c) propose to employ different penalty factors for different weights, enabling large regularization.

**Importance-based pruning**. Importance-based pruning tries to establish certain advanced importance criteria that can reflect the true relative importance among weights as faithfully as possible. The pruned weights are usually decided immediately by some proposed formula instead of by training (although the whole pruning process can involve training, e.g., iterative pruning). The most widely used criterion is the magnitude-based: weight absolute value for unstructured pruningHan et al. (2015; 2016) or $L_1/L_2$-norm for structured pruning Li et al. (2017). This heuristic criterion was proposed a long time ago Reed (1993) and has been argued to be inaccurate. In this respect, improvement mainly comes from using Hessian information to obtain a more accurate approximation of the increased loss when a weight is removed LeCun et al. (1990); Hassibi & Stork (1993). Hessian is intractable to compute for large networks, so some methods (e.g., EigenDamage Wang et al. (2019a), WoodFisher Singh & Alistarh (2020)) employ cheap approximation (such as K-FAC Fisher Martens & Grosse (2015)) to make the 2nd-order criteria tractable on deep networks.

Note that, there is no a hard boundary between the importance-based and regularization-based. Many papers present their schemes in the combination of the two Ding et al. (2018); Wang et al. (2019c). The difference mainly lies in their emphasis: Regularization-based method focuses more on an advanced penalty scheme so that the subsequent pruning criterion can be simple; while the importance-based one focus more on an advanced importance criterion itself. Meanwhile, regularization paradigm always involves iterative training, while the importance-based can be one-shot LeCun et al. (1990); Hassibi & Stork (1993); Wang et al. (2019a) (no training for picking weights to prune) or involve iterative training Molchanov et al. (2017b; 2019); Ding et al. (2019a;b).

**Other model compression methods**. Apart from pruning, there are also many other model compression approaches, e.g., quantization Courbariaux & Bengio (2016); Courbariaux et al. (2016); Rastegari et al. (2016), knowledge distillation Buciluǎ et al. (2006); Hinton et al. (2014), low-

---

[1]By pruning schedule, we mean the way to remove weights (e.g., removing all weights in a single step or multi-steps), *not* the training schedule such as learning rate settings, etc.

rank decomposition Denton et al. (2014); Jaderberg et al. (2014); Lebedev et al. (2014); Zhang et al. (2015), and efficient architecture design or search Howard et al. (2017); Sandler et al. (2018); Howard et al. (2019); Zhang et al. (2018); Tan & Le (2019); Zoph & Le (2017); Elsken et al. (2019). They are orthogonal to network pruning and can work with the proposed methods to compress more.

# 3 PROPOSED METHOD

## 3.1 PROBLEM FORMULATION

Pruning can be formulated as a transformation $T(*)$ that takes a pretrained big model $\mathbf{w}$ as input and output a small model $\mathbf{w}_1$, typically followed by a fine-tuning process $F(*)$, which gives us the final output $\mathbf{w}_2 = F(\mathbf{w}_1)$. We do not focus on $F(*)$ since it is simply a standard neural network training process, but focus on the process of $\mathbf{w}_1 = T(\mathbf{w})$. The effect of pruning can be further specified into two sub-transformations: (1) $M = T_1(\mathbf{w})$, which obtains a binary mask vector $M$ that decides which weights will be removed; (2) $T_2(\mathbf{w})$, which adjusts the values of remaining weights. That is,

$$\mathbf{w}_1 = T(\mathbf{w}) = T_1(\mathbf{w}) \odot T_2(\mathbf{w}) = M \odot T_2(\mathbf{w}). \tag{1}$$

For one-shot pruning, there is no iterative training at $T_1$. It depends on a specific algorithm to decide whether to adjust the remaining weights. For example, OBD LeCun et al. (1990) and $L_1$-norm pruning Li et al. (2017) do not adjust the kept weights (i.e., $T_2$ is the identity function) while OBS Hassibi & Stork (1993) does. For learning-based pruning, both $T_1$ and $T_2$ involve iterative training and the kept weights will always be adjusted.

In the following, we will present our algorithms in the **filter pruning** scenario since we mainly focus on model acceleration instead of compression in this work. Nevertheless, the methodology can seamlessly translate to the unstructured pruning case. The difference lies in how we define the *weight group*: For filter pruning, a 4-d tensor convolutional filter (or 2-d tensor for fully-connected layers) is regarded as a weight group, while for unstructured pruning, a single weight makes a group.

## 3.2 PRUNING SCHEDULE: GREG-1

Our first method (GReg-1) is a variant of $L_1$-norm pruning Li et al. (2017). It obtains the mask $M$ by $L_1$-norm sorting but *adjusts the kept weights via regularization*. Specifically, given a pre-trained model $\mathbf{w}$ and layer pruning ratio $r_l$, we sort the filters by $L_1$-norm and set the mask to zero for those with the least norms. Then, unlike Li et al. (2017) which removes the unimportant weights *immediately* (i.e., one-shot fashion), we impose a growing $L_2$ penalty to drive them to zero first:

$$\lambda_j = \lambda_j + \delta\lambda, j \in \{j \mid M[j] = 0\}, \tag{2}$$

where $\lambda_j$ is the penalty factor for $j$-th weight; $\delta\lambda$ is the granularity in which we add up the penalty. Clearly, a *smaller* $\delta\lambda$ means this regularization process *smoother*. Besides, $\lambda_j$ is only updated every $K_u$ iterations, which is a buffer time to let the network adapt to the new regularization. This algorithm is to explore whether the way we remove them (i.e., pruning schedule) leads to a difference given the *same* weights to prune. Simple as it is, the scheme can bring significant accuracy gains especially under a large pruning ratio (Tab. 1). Note that, we intentionally set $\delta\lambda$ the *same* for all the unimportant weights to keep the core idea simple. Natural extensions of using different penalty factors for different weights (such as those in Ding et al. (2018); Wang et al. (2019c)) may be worth exploring but out of the scope of this work.

When $\lambda_j$ reaches a pre-set ceiling $\tau$, we terminate the training and prune those with the least $L_1$-norms, then fine-tune. Notably, the pruning will *barely* hurt the accuracy since the unimportant weights have been compressed to typically less than $\frac{1}{1000}$ the magnitude of remaining weights.

## 3.3 IMPORTANCE CRITERION: GREG-2

Our second algorithm is to further take advantage of the growing regularization scheme, not for pruning schedule but scoring. The training of neural networks is prone to overfitting, so regularization is normally employed. $L_2$ regularization (or referred to as weight decay) is a standard technique for deep network training. Given a dataset $\mathcal{D}$, model parameters $\mathbf{w}$, the total loss will typically be

$$\mathcal{E}(\mathbf{w}, \mathcal{D}) = \mathcal{L}(\mathbf{w}, \mathcal{D}) + \frac{1}{2}\lambda\|\mathbf{w}\|_2^2, \tag{3}$$

where $\mathcal{L}$ is the task loss function. When the training converges, there should be

$$\lambda w_i^* + \frac{\partial \mathcal{L}}{\partial w_i}\Big|_{w_i=w_i^*} = 0, \tag{4}$$

where $w_i^*$ indicates the $i$-th weight *at its local minimum*. Eq. (4) shows that, for each specific weight element, its equilibrium position is determined by two forces: loss gradient (i.e., guidance from the task) and regularization gradient (i.e., guidance from our prior). Our idea is to slightly increase the $\lambda$ to break the equilibrium and see how it results in a new one. A general impression is: If $\lambda$ goes a little higher, the penalty force will drive the weights further towards origin and it will not stop unless proper loss gradient comes to halt it and then a new equilibrium is reached at $\hat{w}_i^*$. Considering different weights have different scales, we define a ratio $r_i = \hat{w}_i^*/w_i^*$ to describe how much the weight magnitude changes after increasing the penalty factor. Our interest lies in how the $r_i$ differs from one another and how it relates to the underlying Hessian information.

Deep neural networks are well-known over-parameterized and highly non-convex. To obtain a feasible analysis, we adopt a local quadratic approximation of the loss function based on Taylor series expansion Strang (1991) following common practices LeCun et al. (1990); Hassibi & Stork (1993); Wang et al. (2019a). Then when the model is converged, the error $\mathcal{E}$ can be described by the converged weights $\mathbf{w}^*$ and the underlying Hessian matrix $\mathbf{H}$ (note $\mathbf{H}$ is p.s.d. since the model is converged). After increasing the penalty $\lambda$ by $\delta\lambda$, the new converged weights can be proved to be

$$\hat{\mathbf{w}}^* = (\mathbf{H} + \delta\lambda\,\mathbf{I})^{-1}\mathbf{H}\mathbf{w}^*, \tag{5}$$

where $\mathbf{I}$ stands for the identity matrix. Here we meet with the common problem of estimating Hessian and its inverse, which are well-known to be intractable for deep neural networks. We explore two simplified cases to help us move forward.

(1) $\mathbf{H}$ is diagonal, which is a common simplification for Hessian LeCun et al. (1990), implying that the weights are independent of each other. For $w_i^*$ with second derivative $h_{ii}$. With $L_2$ penalty increased by $\delta\lambda$ ($\delta\lambda > 0$), the new converged weights can be proved to be

$$\hat{w}_i^* = \frac{h_{ii}}{h_{ii}+\delta\lambda}w_i^*, \Rightarrow r_i = \frac{\hat{w}_i^*}{w_i^*} = \frac{1}{\delta\lambda/h_{ii}+1}, \tag{6}$$

where $r_i \in [0,1)$ since $h_{ii} \geq 0$ and $\delta\lambda > 0$. As seen, **larger** $h_{ii}$ **results in larger** $r_i$ **(closer to 1), meaning that the weight is relatively** *less* **moved towards the origin**. Our second algorithm primarily builds upon this finding, which implies when we add a penalty perturbation to the converged network, the way that different weights respond can reflect their underlying Hessian information.

(2) In practice, we know $\mathbf{H}$ is rarely diagonal. How the dependency among weights affects the finding above is of interest. To have a closed form of inverse Hessian in Eq. (5), we explore the 2-d case, namely, $\mathbf{w}^* = \left(\begin{smallmatrix} w_1^* \\ w_2^* \end{smallmatrix}\right)$, $\mathbf{H} = \left(\begin{smallmatrix} h_{11} & h_{12} \\ h_{12} & h_{22} \end{smallmatrix}\right)$, $\hat{\mathbf{H}} = \left(\begin{smallmatrix} h_{11}+\delta\lambda & h_{12} \\ h_{12} & h_{22}+\delta\lambda \end{smallmatrix}\right)$. The new converged weights can be analytically solved below, where the approximation equality is because that $\delta\lambda$ is rather small,

$$\left\{\begin{matrix}\hat{w}_1^* \\ \hat{w}_2^*\end{matrix}\right\} = \frac{1}{|\hat{\mathbf{H}}|}\left\{\begin{matrix}(h_{11}h_{22}+h_{11}\delta\lambda-h_{12}^2)w_1^*+\delta\lambda h_{12}w_2^* \\ (h_{11}h_{22}+h_{22}\delta\lambda-h_{12}^2)w_2^*+\delta\lambda h_{12}w_1^*\end{matrix}\right\} \approx \frac{1}{|\hat{\mathbf{H}}|}\left\{\begin{matrix}(h_{11}h_{22}+h_{11}\delta\lambda-h_{12}^2)w_1^* \\ (h_{11}h_{22}+h_{22}\delta\lambda-h_{12}^2)w_2^*\end{matrix}\right\}, \tag{7}$$

$$\Rightarrow r_1 = \frac{1}{|\hat{\mathbf{H}}|}(h_{11}h_{22}+h_{11}\delta\lambda-h_{12}^2), \quad r_2 = \frac{1}{|\hat{\mathbf{H}}|}(h_{11}h_{22}+h_{22}\delta\lambda-h_{12}^2). \tag{8}$$

As seen, $h_{11} > h_{22}$ also leads to $r_1 > r_2$, in line with the finding above. The existence of weight dependency (i.e., the $h_{12}$) actually does *not* affect the conclusion since it is included in both ratios.

These theoretical analyses show us that when the penalty is increased at the same pace, because of different local curvature structures, the weights actually respond differently – weights with **larger** curvature will be **less** moved. As such, the magnitude discrepancy among weights will be magnified as $\lambda$ grows. Ultimately, the weights will naturally separate (see Fig. 1 for an empirical validation). When the discrepancy is large enough, even the simple $L_1$-norm can make an accurate criterion. Notably, the whole process happens itself with the uniformly rising $L_2$ penalty, *no need to know the Hessian values*, thus not bothered by any issue arising from Hessian approximation in relevant prior arts LeCun et al. (1990); Hassibi & Stork (1993); Wang et al. (2019a); Singh & Alistarh (2020).

In terms of the specific algorithm, *all* the penalty factor is increased at the same pace,

$$\lambda_j = \lambda_j + \delta\lambda, \text{ for all } j. \tag{9}$$

---

**Algorithm 1** GReg-1 and GReg-2 Algorithms

---

1: **Input**: Pre-trained model $\mathbf{w}$, pruning ratio for $l$-th layer $r_l, l = 1 \sim L$, original weight decay $\gamma$.
2: **Input**: Regularization ceiling $\tau$, ceiling for picking $\tau'$, interval $K_u, K_s$, granularity $\delta\lambda$.
3: **Init**: Iteration $i = 0$. $\lambda_j = 0$ for all filter $j$. Set kept filter indexes $S_l^k$ to $\varnothing$ for each layer $l$.
4: **Init**: Set pruned filter indexes $S_l^p$ by $L_1$-norm sorting, set $S_l^p$ to full set, for each layer $l$.
5: **while** $\lambda_j \leq \tau, j \in S_l^p$ **do**
6:     **if** $i \% K_u = 0$ **then**
7:         **if** $S_l^k = \varnothing$ and $\lambda_j > \tau', j \in S_l^p$ **then**
8:             Set $S_l^p$ by $L_1$-norm scoring, $S_l^k$ as the complementary set of $S_l^p$, for each layer $l$.
9:         **end if**
10:         $\lambda_j = \lambda_j + \delta\lambda$ for $j \in S_l^p$, $\lambda_j = -\gamma$ for $j \in S_l^k$, for each layer $l$
11:     **end if**
12:     Weight update by stochastic gradient descent (where the regularization is enforced).
13:     $i = i + 1$.
14: **end while**
15: Train for another $K_s$ iterations to stabilize. Then prune by $L_1$-norms and get model $\mathbf{w}_1$.
16: Fine-tune $\mathbf{w}_1$ to regain accuracy.
17: **Output**: Pruned model $\mathbf{w}_2$.

---

When $\lambda_j$ reaches some ceiling $\tau'$, the magnitude gap turns large enough to let $L_1$-norm do scoring faithfully. After this, the procedures are similar to those in GReg-1: $\lambda$ for the unimportant weights are further increased. One extra step is to bring back the kept weights to the normal magnitude. Although they are the "survivors" during the previous competition under a large penalty, their expressivity are also hurt. To be exact, we adopt *negative* penalty factor for the kept weights to encourage them to recover. When the $\lambda$ for unimportant weights reaches the threshold $\tau$ (akin to that of GReg-1), the training is terminated. $L_1$-pruning is conducted and then fine-tune to regain accuracy. To this end, the proposed two algorithms can be summarized in Algorithm 1.

**Pruning ratios**. We employ *pre-specified* pruning ratios in this work to keep the core method neat (see Appendix for more discussion). Exploring layer-wise sensitivity is out of the scope of this work, but clearly any method that finds more proper pruning ratios can readily work with our approaches.

**Discussion: differences from IncReg**. Although our work shares a general spirit of growing regularization with IncReg Wang et al. (2019c;b), our work is actually starkly different from theirs in many axes:

- Motivation. The motivations for using the growing regularization are different. Wang et al. (2019c;b) adopt growing regularization to select the unimportant weights by training. Namely, they focus on the importance criterion problem. In contrast, we use growing regularization to investigate the pruning schedule problem (for GReg-1) or exploit the underlying Hessian information (for GReg-2). The importance criterion is simply $L_1$-norm.

- Algorithm design. Wang et al. (2019c;b) assign different regularization factors to different weight groups based on their relative importance, while we assign them with the same factors. For GReg-1, this may not be a substantial difference, while for GReg-2, the difference is fundamental because the theoretical analysis of GReg-2 (Sec. 3.3) relies on the fact that regularization factors are kept the same for different weights.

- Theoretical analysis. The algorithm in Wang et al. (2019c;b) is generally heuristic-based, while our work provides rigorous theoretical analyses (Sec. 3.3) to support the proposed algorithm GReg-2.

- Empirical performance. Both our methods are significantly better than Wang et al. (2019c;b) on the large-scale ImageNet dataset, which will be shown in the experiment section (Tab. 3).

**Discussion: other regularization forms**. The proposed methods in this work adopts $L_2$ regularization. Here we discuss the possibility to generalize the method to other regularization forms ($L1$ and $L_0$). (1) For GReg-1, it can be easily generalized to other regularization forms like $L_1$. For GReg-2, since the theoretical basis in Sec. 3.3 relies on the local quadratic approximation, $L_2$ regularization meets this requirement while $L_1$ does not. Therefore, GReg-2 cannot be (easily) generalized to the

Table 1: Comparison between pruning schedules: one-shot pruning vs. our proposed GReg-1. Each setting is randomly run for 3 times, mean and std accuracies reported.

| **ResNet56 + CIFAR10**: Baseline accuracy 93.36%, #Params: 0.8530M, FLOPs: 0.1255G | | | | | |
|---|---|---|---|---|---|
| Pruning ratio $r$ (%) | 50 | 70 | 90 | 92.5 | 95 |
| Sparsity (%) / Speedup | 49.82/1.99$\times$ | 70.57/3.59$\times$ | 90.39/11.41$\times$ | 93.43/14.76$\times$ | 95.19/19.31$\times$ |
| Acc. (%, $L_1$+one-shot) | $92.97_{\pm 0.15}$ | $91.88_{\pm 0.09}$ | $87.34_{\pm 0.21}$ | $87.31_{\pm 0.28}$ | $82.79_{\pm 0.22}$ |
| Acc. (%, GReg-1, ours) | $\mathbf{93.06_{\pm 0.09}}$ | $\mathbf{92.23_{\pm 0.21}}$ | $\mathbf{89.49_{\pm 0.23}}$ | $\mathbf{88.39_{\pm 0.15}}$ | $\mathbf{85.97_{\pm 0.16}}$ |
| Acc. gain (%) | 0.09 | 0.35 | 2.15 | 1.08 | 3.18 |
| **VGG19 + CIFAR100**: Baseline accuracy 74.02%, #Params: 20.0812M, FLOPs: 0.3982G | | | | | |
| Pruning ratio $r$ (%) | 50 | 60 | 70 | 80 | 90 |
| Sparsity (%) / Speedup | 74.87/3.60$\times$ | 84.00/5.41$\times$ | 90.98/8.84$\times$ | 95.95/17.30$\times$ | 98.96/44.22$\times$ |
| Acc. (%, $L_1$+one-shot) | $71.49_{\pm 0.14}$ | $70.27_{\pm 0.12}$ | $66.05_{\pm 0.04}$ | $61.59_{\pm 0.03}$ | $51.36_{\pm 0.11}$ |
| Acc. (%, GReg-1, ours) | $\mathbf{71.50_{\pm 0.12}}$ | $\mathbf{70.33_{\pm 0.12}}$ | $\mathbf{67.35_{\pm 0.15}}$ | $\mathbf{63.55_{\pm 0.29}}$ | $\mathbf{57.09_{\pm 0.03}}$ |
| Acc. gain (%) | 0.01 | 0.06 | 1.30 | 1.96 | 5.73 |

$L_1$ regularization as far as we can see currently. (2) For $L_0$ regularization, it is well-known NP-hard. In practice, it is typically converted to the $L_1$ regularization case, which we just discussed.

## 4 EXPERIMENTAL RESULTS

**Datasets and networks**. We first conduct analyses on the CIFAR10/100 datasets Krizhevsky (2009) with ResNet56 He et al. (2016)/VGG19 Simonyan & Zisserman (2015). Then we evaluate our methods on the large-scale ImageNet dataset Deng et al. (2009) with ResNet34 and 50 He et al. (2016). For CIFAR datasets, we train our baseline models with accuracies comparable to those in the original papers. For ImageNet, we take the official PyTorch Paszke et al. (2019) pre-trained models[2] as baseline to maintain comparability with other methods.

**Training settings**. To control the irrelevant factors as we can, for comparison methods that release their pruning ratios, we will adopt their ratios; otherwise, we will use our specified ones. We compare the speedup (measured by FLOPs reduction) since we mainly target model acceleration rather than compression. Detailed training settings (e.g., hyper-parameters and layer pruning ratios) are summarized in the Appendix.

### 4.1 RESNET56/VGG19 ON CIFAR-10/100

**Pruning schedule: GReg-1**. First, we explore the effect of different pruning schedules on the performance of pruning. Specifically, we conduct two sets of experiments for comparison: (1) prune by $L_1$-norm sorting and fine-tune Li et al. (2017) (shorted as "$L_1$+one-shot"); (2) employ the proposed growing regularization scheme ("GReg-1") and fine-tune. We use a uniform pruning ratio scheme here: Pruning ratio $r$ is the *same* for all $l$-th conv layer (the first layer is *not* pruned following common practice Gale et al. (2019)). For ResNet56, since it has the residual addition restriction, we only prune the first conv layer in a block as previous works do Li et al. (2017). For comprehensive comparisons, the pruning ratios vary in a large spectrum, covering acceleration ratios from around $2\times$ to $44\times$. Note that we do not intend to obtain the best performance here but systematically explore the effect of different pruning schedules, so we employ relatively simple settings (e.g., the uniform pruning ratios). For fair comparisons, the fine-tuning scheme (e.g., number of epochs, learning rate schedule, etc.) is *the same* for different methods. Therefore, the key comparison here is to see which method can deliver a better base model before fine-tuning.

The results are shown in Tab. 1. We have the following observations: (1) On the whole, the proposed GReg-1 *consistently* outperforms $L_1$+one-shot. It is important to reiterate that the two settings have *exactly the same* pruned weights, so the only difference is how they are removed. The accuracy gaps show that apart from importance scoring, pruning schedule is also a critical factor. In the Appendix D, we present more results to demonstrate this finding actually is *general*, not merely limited to the case of $L_1$-norm criterion. The proposed regularization-based pruning schedule is consistently more favorable than the one-shot counterpart. (2) The larger pruning ratio, the more

---

[2]https://pytorch.org/docs/stable/torchvision/models.html

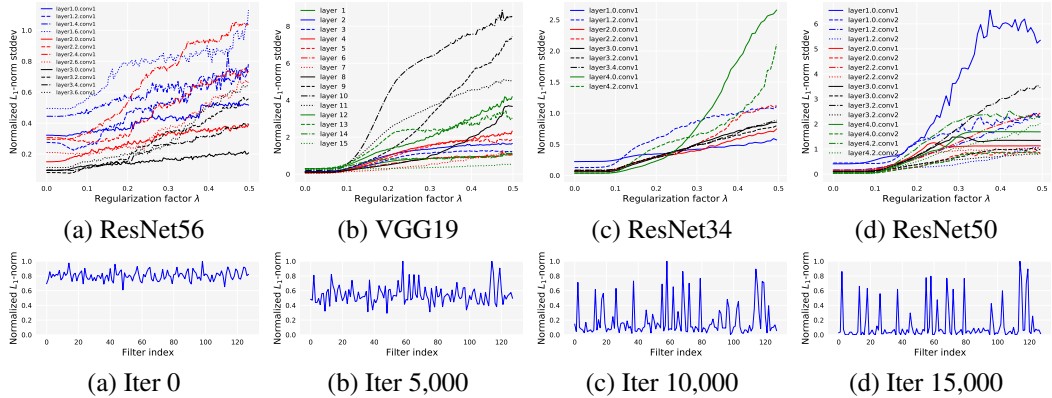

(a) ResNet56     (b) VGG19     (c) ResNet34     (d) ResNet50

(a) Iter 0     (b) Iter 5,000     (c) Iter 10,000     (d) Iter 15,000

Figure 1: Row 1: Illustration of weight separation as $L_2$ penalty grows. Row 2: Normalized filter $L_1$-norm over iterations for ResNet50 layer2.3.conv1 (please see the Appendix for VGG19 plots).

Table 2: Comparison of different methods on the CIFAR10 and CIFAR100 datasets.

| Method | Network/Dataset | Base acc. (%) | Pruned acc. (%) | Acc. drop | Speedup |
|---|---|---|---|---|---|
| CP He et al. (2017) | | 92.80 | 91.80 | 1.00 | 2.00× |
| AMC He et al. (2018b) | | 92.80 | 91.90 | 0.90 | 2.00× |
| SFP He et al. (2018a) | ResNet56/CIFAR10 | 93.59 | 93.36 | 0.23 | 2.11× |
| AFP Ding et al. (2018) | | 93.93 | 92.94 | 0.99 | **2.56×** |
| C-SGD Ding et al. (2019a) | | 93.39 | **93.44** | -0.05 | 2.55× |
| GReg-1 (**ours**) | | 93.36 | 93.18 | 0.18 | 2.55× |
| GReg-2 (**ours**) | | 93.36 | 93.36 | 0.00 | 2.55× |
| Kron-OBD Wang et al. (2019a) | | 73.34 | 60.70 | 12.64 | 5.73× |
| Kron-OBS Wang et al. (2019a) | | 73.34 | 60.66 | 12.68 | 6.09× |
| EigenDamage Wang et al. (2019a) | VGG19/CIFAR100 | 73.34 | 65.18 | 8.16 | 8.80× |
| GReg-1 (**ours**) | | 74.02 | 67.55 | 6.67 | **8.84×** |
| GReg-2 (**ours**) | | 74.02 | **67.75** | **6.47** | **8.84×** |

pronounced of the gain. This is reasonable since when more weights are pruned, the network cannot recover by its inherent plasticity Mittal et al. (2018), then the regularization-based way is more helpful because it helps the model transfer its expressive power to the remaining part. When the pruning ratio is relatively small (such as ResNet56, $r = 50\%$), the plasticity of the model is enough to heal, so the benefit from GReg-1 is less significant compared with the one-shot counterpart.

**Importance criterion: GReg-2.** Here we empirically validate our finding in Sec. 3.3, that is, with uniformly rising $L_2$ penalty, the weights should naturally separate. We claim, if $h_{11} > h_{22}$, there should be $r_1 > r_2$, where $r_1 = \frac{\hat{w_1}}{w_1}, r_2 = \frac{\hat{w_2}}{w_2}$ (the * mark indicating the local minimum is omitted here for readability). $r_1 > r_2$ leads to $\frac{\hat{w_1}}{w_1} > \frac{\hat{w_2}}{w_2}$, namely, $r_1 = \frac{\hat{w_1}}{\hat{w_2}} > \frac{w_1}{w_2}$. This shows that, after the $L_2$ penalty grows a little, the new magnitude ratio of weight 1 over weight 2 will be magnified if $h_{11} > h_{22}$ ($w_1, w_2$ are positive in the analysis here, while the conclusion still holds if either of them is negative). In Fig. 1 (Row 1), we plot the standard deviation (divided by the means for normalization since the magnitude varies over iterations) of filter $L_1$-norms as the regularization grows. As seen, the normalized $L_1$-norm stddev grows larger and larger as $\lambda$ grows. This phenomenon *consistently* appears across different models and datasets. To figuratively understand how the increasing penalty affects the relative magnitude over time, in Fig. 1 (Row 2), we plot the relative $L_1$-norms (divided by the max $L_1$-norm for normalization) at different iterations. As shown, it is hard to tell which filters are really important by the initial filter magnitude (Iter 0), but under a large penalty later, their discrepancy turns more and more obvious and finally it is very easy to identify which filters are more important. Since the magnitude gap is so large, the simple $L_1$-norm can make a sufficiently faithful criterion.

**CIFAR benchmarks**. Finally, we compare the proposed algorithms with existing methods on the CIFAR datasets (Tab. 2). Here we adopt non-uniform pruning ratios (see the Appendix for specific numbers) for the best accuracy-FLOPs trade-off. On CIFAR10, compared with AMC He et al.

Table 3: Acceleration comparison on ImageNet. FLOPs: ResNet34: 3.66G, ResNet50: 4.09G.

| Method | Network | Base top-1 (%) | Pruned top-1 (%) | Top-1 drop | Speedup |
|---|---|---|---|---|---|
| $L_1$ (pruned-B) Li et al. (2017) | | 73.23 | 72.17 | 1.06 | **1.32×** |
| Taylor-FO Molchanov et al. (2019) | ResNet34 | 73.31 | 72.83 | 0.48 | 1.29× |
| GReg-1 (**ours**) | | 73.31 | 73.54 | -0.23 | **1.32×** |
| GReg-2 (**ours**) | | 73.31 | **73.61** | **-0.30** | **1.32×** |
| ProvableFP Liebenwein et al. (2020) | ResNet50 | 76.13 | 75.21 | 0.92 | 1.43× |
| GReg-1 (**ours**) | | 76.13 | **76.27** | **-0.14** | **1.49×** |
| AOFP Ding et al. (2019b) | ResNet50 | 75.34 | 75.63 | -0.29 | **1.49×** |
| GReg-1 (**ours**)* | | 75.40 | **76.13** | **-0.73** | **1.49×** |
| IncReg Wang et al. (2019b) | | 75.60 | 72.47 | 3.13 | 2.00× |
| SFP He et al. (2018a) | | 76.15 | 74.61 | 1.54 | 1.72× |
| HRank Lin et al. (2020a) | | 76.15 | 74.98 | 1.17 | 1.78× |
| Taylor-FO Molchanov et al. (2019) | | 76.18 | 74.50 | 1.68 | 1.82× |
| Factorized Li et al. (2019) | ResNet50 | 76.15 | 74.55 | 1.60 | **2.33×** |
| DCP Zhuang et al. (2018) | | 76.01 | 74.95 | 1.06 | 2.25× |
| CCP-AC Peng et al. (2019) | | 76.15 | 75.32 | 0.83 | 2.18× |
| GReg-1 (**ours**) | | 76.13 | 75.16 | 0.97 | 2.31× |
| GReg-2 (**ours**) | | 76.13 | **75.36** | **0.77** | 2.31× |
| C-SGD-50 Ding et al. (2019a) | | 75.34 | 74.54 | 0.80 | 2.26× |
| AOFP Ding et al. (2019b) | ResNet50 | 75.34 | 75.11 | 0.23 | **2.31×** |
| GReg-2 (**ours**)* | | 75.40 | **75.22** | **0.18** | **2.31×** |
| LFPC He et al. (2020) | | 76.15 | 74.46 | 1.69 | 2.55× |
| GReg-1 (**ours**) | ResNet50 | 76.13 | 74.85 | 1.28 | **2.56×** |
| GReg-2 (**ours**) | | 76.13 | **74.93** | **1.20** | **2.56×** |
| IncReg Wang et al. (2019b) | | 75.60 | 71.07 | 4.53 | 3.00× |
| Taylor-FO Molchanov et al. (2019) | ResNet50 | 76.18 | 71.69 | 4.49 | 3.05× |
| GReg-1 (**ours**) | | 76.13 | 73.75 | 2.38 | **3.06×** |
| GReg-2 (**ours**) | | 76.13 | **73.90** | **2.23** | **3.06×** |

\* Since the base models of C-SGD and AOFP have a much lower accuracy than ours, for fair comparison, we train our own base models with similar accuracy.

Table 4: Compression comparison on ImageNet with ResNet50. #Parameters: 25.56M.

| Method | Base top-1 (%) | Pruned top-1 (%) | Top-1 drop | Sparsity (%) |
|---|---|---|---|---|
| GSM Ding et al. (2019c) | 75.72 | 74.30 | 1.42 | 80.00 |
| Variational Dropout Molchanov et al. (2017a) | 76.69 | 75.28 | 1.41 | 80.00 |
| DPF Lin et al. (2020b) | 75.95 | 74.55 | 1.40 | 82.60 |
| WoodFisher Singh & Alistarh (2020) | 75.98 | 75.20 | 0.78 | **82.70** |
| GReg-1 (**ours**) | 76.13 | **75.45** | **0.68** | **82.70** |
| GReg-2 (**ours**) | 76.13 | 75.27 | 0.86 | **82.70** |

(2018b), though it adopts better layer-wise pruning ratios via reinforcement-learning, our algorithms can still deliver more favorable performance using sub-optimal human-specified ratios. AFP Ding et al. (2018) is another work exploring large regularization, while they do not adopt the *growing* scheme as we do. Its performance is also less favorable on CIFAR10 as shown in the table. Although our methods perform a little worse than C-SGD Ding et al. (2019a) on CIFAR10, on the large-scale ImageNet dataset, we will show our methods are significantly better than C-SGD.

Notably, on CIFAR100, Kron-OBD/OBS (an extension by Wang et al. (2019a) of the original OBD/OBS from unstructured pruning to structured pruning) are believed to be more accurate than $L_1$-norm in terms of capturing relative weight importance LeCun et al. (1990); Hassibi & Stork (1993); Wang et al. (2019a). Yet, they are significantly outperformed by our GReg-1 based on the simple $L_1$-norm scoring. This may inspire us that an average pruning schedule (like the one-shot fashion) can offset the gain from a more advanced importance scoring scheme.

## 4.2 ResNet34/50 on ImageNet

Then we evaluate our methods on the standard large-scale ImageNet benchmarks with ResNets He et al. (2016). We refer to the official PyTorch ImageNet training example[3] to make sure the imple-

---

[3]https://github.com/pytorch/examples/tree/master/imagenet

mentation (such as data augmentation, weight decay, momentum, etc.) is standard. Please refer to the summarized training setting in the Appendix for details.

The results are shown in Tab. 3. Methods with similar speedup are grouped together for easy comparison. In general, our method achieves comparable or better performance across various speedups on ResNet34 and 50. Concretely, (1) On both ResNet34 and 50, when the speedup is small (less than $2\times$), only our methods (and AOFP Ding et al. (2019b) for ResNet50) can even *improve* the top-1 accuracy. This phenomenon is broadly found by previous works Wen et al. (2016); Wang et al. (2018); He et al. (2017) but mainly on small datasets like CIFAR, while we make it on the much challenging ImageNet benchmark. (2) Similar to the results on CIFAR (Tab. 1), when the speedup is larger, the advantage of our method is more obvious. For example, ours GReg-2 only outperforms Taylor-FO Molchanov et al. (2019) by 0.86% top-1 accuracy at the $\sim 2\times$ setting, while at $\sim 3\times$, GReg-2 is better by 2.21% top-1 accuracy. (3) Many methods work on the weight importance criterion problem, including some very recent ones (ProvableFP Liebenwein et al. (2020), LFPC He et al. (2020)). Yet as shown, our simple variant of $L_1$-norm pruning can still be a strong competitor in terms of accuracy-FLOPs trade-off. This reiterates one of our key ideas in this work that the pruning schedule may be as important as weight importance scoring and worth more research attention.

**Unstructured pruning**. Although we mainly target filter pruning in this work, the proposed methods actually can be applied to unstructured pruning as effectively. In Tab. 4, we present the results of unstructured pruning on ResNet50. WoodFisher Singh & Alistarh (2020) is the state-of-the-art Hessian-based unstructured pruning approach. Notably, without any Hessian approximation, our GReg-2 can achieve comparable performance with it (better absolute accuracy, yet slightly worse accuracy drop). Besides, the simple magnitude pruning variant GReg-1 delivers more favorable result, implying that a better pruning schedule also matters in the unstructured pruning case.

## 5 CONCLUSION

Regularization is long deemed as a sparsity-learning tool in neural network pruning, which usually works in the small strength regime. In this work, we present two algorithms that exploit regularization in a new fashion that the penalty factor is uniformly raised to a large level. Two central problems regarding deep neural pruning are tackled by the proposed methods, pruning schedule and weight importance criterion. The proposed approaches rely on few impractical assumptions, have a sound theoretical basis, and are scalable to large datasets and networks. Apart from the methodology itself, the encouraging results on CIFAR and ImageNet also justify our general ideas in this paper: (1) In addition to weight importance scoring, pruning schedule is another pivotal factor in deep neural pruning which may deserve more research attention. (2) Without any Hessian approximation, we can still tap into its power for pruning with the help of growing $L_2$ regularization.

## ACKNOWLEDGEMENTS

The work is supported by the National Science Foundation Award ECCS-1916839 and the U.S. Army Research Office Award W911NF-17-1-0367.

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

# A APPENDIX

## A.1 EXPERIMENTAL SETTING DETAILS

**Training setting summary**. About the networks evaluated, we intentionally avoid AlexNet and VGG on the ImageNet benchmark because the single-branch architecture is no longer representative of the modern deep network architectures with residuals (but still keep VGG19 on the CIFAR analysis to make sure the findings are not limited to one specific architecture). Apart from some key settings stated in the paper, a more detailed training setting summary is shown as Tab. 5.

Table 5: Training setting summary. For the SGD solver, in the parentheses are the momentum and weight decay. For ImageNet, batch size 64 is used for pruning instead of the standard 256, which is because we want to save the training time.

| Dataset | CIFAR | ImageNet |
|---|---|---|
| Solver | SGD (0.9, 5e-4) | SGD (0.9, 1e-4) |
| LR policy (prune) | Fixed (1e-3) | |
| LR policy (finetune) | Multi-step (0:1e-2, 60:1e-3, 90:1e-4) | Multi-step (0:1e-2, 30:1e-3, 60:1e-4, 75:1e-5) |
| Total epoch (finetune) | 120 | 90 |
| Batch size (prune) | 256 | 64 |
| Batch size (finetune) | 256 | |

**Pruning ratios**. Although several recent methods Ding et al. (2019b); Singh & Alistarh (2020) can automatically decide pruning ratios, in this paper we opt to consider pruning *independent* with the pruning ratio choosing. The main consideration is that pruning ratio is broadly believed to reflect the redundancy of different layers LeCun et al. (1990); Wen et al. (2016); He et al. (2017), which is an *inherent characteristic of the model*, thus should not be coupled with the subsequent pruning algorithms.

Before we list the specific pruning ratios, we explain how we set them. **(1)** For a ResNet, if it has $N$ stages, we will use a list of $N$ floats to represent its pruning ratios for the $N$ stages. For example, ResNet56 has 4 stages in conv layers, then "[0, 0.5, 0.5, 0.5]" means "for the first stage (which is also the first conv layer), the pruning ratio is 0; the other three stages have pruning ratio of 0.5". Besides, since we do not prune the last conv in a residual block, which means for a two-layer residual block (for ResNet56), we only prune the first layer; for a three-layer bottleneck block (for ResNet34 and 50), we only prune the first and second layers. **(2)** For VGG19, we use the following pruning ratio setting. For example, "[0:0, 1-9:0.3, 10-15:0.5]" means "for the first layer (index starting from 0), the pruning ratio is 0; for layer 1 to 9, the pruning ratio is 0.3; for layer 10 to 15, the pruning ratio is 0.5".

With these, the specific pruning ratio for each of our experiments in the paper are listed in Tab. 6. We do not have strong rules to set them, except one, which is setting the pruning ratios of *higher* stages *smaller*, because the FLOPs of higher layers are relatively smaller (due to the fact that the spatial feature map sizes are smaller) and we are targeting more acceleration than compression. Of course, this scheme only is quite crude, yet as our results (Tab. 3 and 4) show, even with these crude settings, the performances are still competitive.

# B PROOF OF EQ. 5

When a quadratic function $\mathcal{E}$ converges at $\mathbf{w}^*$ with Hessian matrix $\mathbf{H}$, it can be formulated as

$$\mathcal{E} = (\mathbf{w} - \mathbf{w}^*)^T \mathbf{H}(\mathbf{w} - \mathbf{w}^*) + C, \tag{10}$$

where $C$ is a constant. Now a new function is made by increasing the $L_2$ penalty by small amount $\delta\lambda$, namely,

$$\hat{\mathcal{E}} = \mathcal{E} + \delta\lambda\mathbf{w}^T\mathbf{I}\mathbf{w}. \tag{11}$$

Let the new converged values be $\hat{\mathbf{w}}^*$, then similar to Eq. 10, $\hat{\mathcal{E}}$ can be formulated as

$$\hat{\mathcal{E}} = (\mathbf{w} - \hat{\mathbf{w}}^*)^T \hat{\mathbf{H}}(\mathbf{w} - \hat{\mathbf{w}}^*) + \hat{C}, \text{ where } \hat{\mathbf{H}} = \mathbf{H} + \delta\lambda\mathbf{I}. \tag{12}$$

Table 6: Pruning ratio summary.

| Dataset | Network | Speedup | Pruned top-1 accuracy (%) | Pruning ratio |
|---|---|---|---|---|
| CIFAR10 | ResNet56 | $2.55\times$ | 93.36 | [0, 0.75, 0.75, 0.32, 0] |
| CIFAR100 | VGG19 | $8.84\times$ | 67.56 | [1-15:0.7] |
| ImageNet | ResNet34 | $1.32\times$ | 73.44 | [0, 0.50, 0.60, 0.40, 0, 0]* |
| ImageNet | ResNet50 | $1.49\times$ | 76.24 | [0, 0.30, 0.30, 0.30, 0.14, 0] |
| ImageNet | ResNet50 | $2.31\times$ | 75.16 | [0, 0.60, 0.60, 0.60, 0.21, 0] |
| ImageNet | ResNet50 | $2.56\times$ | 74.75 | [0, 0.74, 0.74, 0.60, 0.21, 0] |
| ImageNet | ResNet50 | $3.06\times$ | 73.50 | [0, 0.68, 0.68, 0.68, 0.50, 0] |

* In addition to the pruning ratios, several layers are skipped, following the setting of $L_1$ (pruned-B) Li et al. (2017). Specifically, we refer to the implementation of Liu et al. (2019) at https://github.com/Eric-mingjie/rethinking-network-pruning/tree/master/imagenet/l1-norm-pruning.

Meanwhile, combine Eq. 10 and Eq. 11, we can obtain

$$\hat{\mathcal{E}} = (\mathbf{w} - \mathbf{w}^*)^T \mathbf{H}(\mathbf{w} - \mathbf{w}^*) + \delta\lambda\mathbf{w}^T\mathbf{I}\mathbf{w} + C. \tag{13}$$

Compare Eq. 13 with Eq. 12, we have

$$(\mathbf{H} + \delta\lambda\mathbf{I})\hat{\mathbf{w}}^* = \mathbf{H}\mathbf{w}^* \Rightarrow \hat{\mathbf{w}}^* = (\mathbf{H} + \delta\lambda\mathbf{I})^{-1}\mathbf{H}\mathbf{w}^*. \tag{14}$$

## C   PROOF OF EQ. 7

$$\hat{\mathbf{H}} = \begin{Bmatrix} h_{11} + \delta\lambda & h_{12} \\ h_{12} & h_{22} + \delta\lambda \end{Bmatrix} \Rightarrow \hat{\mathbf{H}}^{-1} = \frac{1}{|\hat{\mathbf{H}}|} \begin{Bmatrix} h_{22} + \delta\lambda & -h_{12} \\ -h_{12} & h_{11} + \delta\lambda \end{Bmatrix} \tag{15}$$

Therefore, $\hat{\mathbf{w}}^* = \hat{\mathbf{H}}^{-1}\mathbf{H}\mathbf{w}^* \Rightarrow$

$$\begin{aligned}
\begin{Bmatrix} \hat{w}_1^* \\ \hat{w}_2^* \end{Bmatrix} = \hat{\mathbf{H}}^{-1}\mathbf{H} \begin{Bmatrix} w_1^* \\ w_2^* \end{Bmatrix} &= \frac{1}{|\hat{\mathbf{H}}|} \begin{Bmatrix} h_{22} + \delta\lambda & -h_{12} \\ -h_{12} & h_{11} + \delta\lambda \end{Bmatrix} \begin{Bmatrix} h_{11} & h_{12} \\ h_{12} & h_{22} \end{Bmatrix} \begin{Bmatrix} w_1^* \\ w_2^* \end{Bmatrix} \\
&= \frac{1}{|\hat{\mathbf{H}}|} \begin{Bmatrix} (h_{11}h_{22} + h_{11}\delta\lambda - h_{12}^2)w_1^* + \delta\lambda h_{12}w_2^* \\ (h_{11}h_{22} + h_{22}\delta\lambda - h_{12}^2)w_2^* + \delta\lambda h_{12}w_1^* \end{Bmatrix}.
\end{aligned} \tag{16}$$

## D   GREG-1 + OBD

In Sec. 4.1, we show when pruning the *same* weights, GReg-1 is significantly better than the one-shot counterpart, where the pruned weights are selected by the $L_1$-norm criterion. Here we conduct the same comparison just with a different pruning criterion introduced in OBD LeCun et al. (1990). OBD is also an one-shot pruning method, using a Hessian-based criterion which is believed to be more advanced than $L_1$-norm.

Results are shown in Tab. 7. As seen, using this more advanced importance criterion, our pruning scheme based on growing regularization is still *consistently better* than the one-shot counterpart. Besides, it is also verified here that a better pruning schedule can bring *more* accuracy gain when the speedup is *larger*.

## E   FILTER L1-NORM CHANGE OF VGG19

In Fig. 1 (Row 2), we plot the filter $L_1$-norm change over time for ResNet50 on ImageNet. Here we plot the case of VGG19 on CIFAR100 to show the weight separation phenomenon under growing regularization is a *general* one across different datasets and networks.

## F   HYPER-PARAMETERS AND SENSITIVITY ANALYSIS

There are five introduced values in our methods: regularization ceiling $\tau$, ceiling for picking $\tau'$, interval $K_u, K_s$, granularity $\delta\lambda$. Their settings are summarized in Tab. 8. Among them, the ceilings

Table 7: Comparison between pruning schedules: one-shot pruning vs. our proposed GReg-1 *using the Hessian-based criterion introduced in OBD LeCun et al. (1990)*. Each setting is randomly run for 3 times, mean and std accuracies reported. We vary the global pruning ratio from 0.7 to 0.95 so as to cover the major speedup spectrum of interest. Same as Tab. 1, the pruned weights here are exactly the same for the two methods under each speedup ratio. The finetuning processes (number of epochs, LR schedules, etc.) are also the same to keep fair comparison.

| ResNet56 + CIFAR10: Baseline accuracy 93.36%, #Params: 0.8530M, FLOPs: 0.1255G | | | | | |
|---|---|---|---|---|---|
| Speedup | 2.15× | 3.00× | 4.86× | 5.80× | 6.87× |
| Acc. (%, OBD) | 92.90 (0.05) | 91.90 (0.04) | 89.82 (0.11) | 88.56 (0.11) | 86.90 (0.03) |
| Acc. (%, Ours) | 92.94 (0.12) | 92.27 (0.14) | 90.37 (0.17) | 89.78 (0.06) | 88.69 (0.06) |
| Acc. gain (%) | 0.04 | 0.37 | 0.55 | 1.22 | 1.79 |
| VGG19 + CIFAR100: Baseline accuracy 74.02%, #Params: 20.0812M, FLOPs: 0.3982G | | | | | |
| Speedup | 1.92× | 2.96× | 5.89× | 7.69× | 11.75× |
| Acc. (%, OBD) | 72.68 (0.08) | 70.42 (0.16) | 62.54 (0.13) | 59.18 (0.32) | 54.19 (0.57) |
| Acc. (%, Ours) | 73.08 (0.11) | 71.30 (0.28) | 65.83 (0.13) | 62.87 (0.20) | 59.53 (0.10) |
| Acc. gain (%) | 0.40 | 0.88 | 3.29 | 3.69 | 5.34 |

(a) Iter 0  (b) Iter 10,000  (c) Iter 20,000  (d) Iter 30,000

Figure 2: Normalized filter $L_1$-norm over iterations for VGG19 layer3.

are set through validation: $\tau = 1$ is set to make sure the unimportant weights are pushed down enough (as stated in the main paper, normally after the regularization training, their magnitudes are too small to cause significant accuracy degradation if they are completely removed). $\tau' = 0.01$ is set generally for the same goal as $\tau$, but since it is applied to *all* the weight (not just the unimportant ones), we only expect it to be moderately large (thus smaller than $\tau$) so that the important and unimportant can be differentiated with a clear boundary. For the $\delta\lambda$, we use a very *small* regularization granularity $\delta\lambda$, which our theoretical analysis is based on. We set its value to 1e-4 for GReg-1 and 1e-5 for GReg-2 with reference to the original weight decay value $5 \times 10^{-4}$ (for CIFAR models) and $10^{-4}$ (for ImageNet models). Note that, these values come from our methods per se, not directly related to datasets and networks, thus are invariant to them. This is why we can employ the same setting of these three hyper-parameters in *all* our experiments, freeing practitioners from heavy tuning when dealing with different networks or datasets.

Table 8: Hyper-parameters of our methods.

| Notation | Default value (CIFAR) | Default value (ImageNet) |
|---|---|---|
| $\delta\lambda$ | GReg-1: 1e-4, GReg-2: 1e-5 | |
| $\tau$ | 1 | |
| $\tau'$ | 0.01 | |
| $K_u$ | 10 iterations | 5 iterations |
| $K_s$ | 5k iterations | 40k iterations |

A little bit of change is for $K_u, K_s$. Both are generally to let the network have enough time to converge to the new equilibrium. Generally, we prefer large update intervals, yet we also need to consider the time complexity: Too large of them will bring too many iterations, which may be unnecessary. Among them, $K_s$ is less important since it is to stabilize the large regularization ($\tau = 1$). We introduce it simply to make sure the training is fully converged. Therefore, the possibly more sensitive hyper-parameter is the $K_u$ (set to 5 for ImageNet and 10 for CIFAR). Here we will show the performance is insensitive to the varying $K_u$. As shown in Tab. 9, the peak performance appears at around $K_u = 15$ for ResNet56 and $K_u = 10$ for VGG19. We simply adopt 10 for a uniform setting in our paper. We did not heavily tune these hyper-parameters, yet as seen, they work pretty well across different networks and datasets. Notably, even for the *worst* cases in Tab. 9 (in

blue color), they are still significantly better than those of the "$L_1$+one-shot" scheme, demonstrating the robustness of the proposed algorithm.

Table 9: Sensitivity analysis of $K_u$ on CIFAR10/100 datasets with the proposed GReg-1 algorithm. $K_u = 10$ is the default setting. Pruning ratio 90% (ResNet56) and 70% (VGG19) are explored here. Experiments are randomly run for 3 times with mean accuracy and standard deviation reported. **The best** is highlighted with **bold** and the worst is highlighted with blue color.

| $K_u$ | 1 | 5 | 10 | 15 | 20 | $L_1$+one-shot |
|---|---|---|---|---|---|---|
| Acc. (%, ResNet56) | $89.40_{\pm0.04}$ | $89.38_{\pm0.13}$ | $89.49_{\pm0.23}$ | $\mathbf{89.69_{\pm0.05}}$ | $89.62_{\pm0.13}$ | $87.34_{\pm0.21}$ |
| Acc. (%, VGG19) | $67.22_{\pm0.33}$ | $67.32_{\pm0.24}$ | $\mathbf{67.35_{\pm0.15}}$ | $67.06_{\pm0.40}$ | $66.93_{\pm0.22}$ | $66.05_{\pm0.04}$ |

## G  MORE RESULTS OF PRUNING SCHEDULE COMPARISON

In Tab. 1, we show using $L_1$-norm sorting, our proposed GReg-1 can consistently surpass the one-shot schedule even pruning the same weights. Here we ask a more general question: Can the benefits from a regularization-based schedule consistently appear, *agnostic to the weight importance scoring criterion*? This question is important because it will show if the gain from a better pruning schedule is only a bonus concurrent with the $L_1$ criterion or a really universal phenomenon. Since there are literally so many weight importance criteria, we cannot ablate them one by one. Nevertheless, given a pre-trained model and a pruning ratio $r$, no matter what criterion, its role is to select a filter *subset*. For example, if there are 100 filters in a layer and $r = 0.5$, then they are at most $\binom{100}{50}$ importance criteria in theory for this layer. We can simply *randomly* pick a subset of filters (which corresponds to certain criterion, albeit unknown) and compare the one-shot way with regularization-based way on the subset. Based on this idea, we conduct five random runs on the ResNet56 and VGG19 to explore this. The pruning ratio is chosen as 90% for ResNet56 and 70% for VGG19 because under this ratio the compression (or acceleration) ratio is about 10 times, neither too large nor too small (where the network can heal itself regardless of pruning methods).

The results are shown in Tab. 10. Here is a sanity check: Compared with Tab. 1, the mean accuracy of pruning randomly picked filters should be *less* than pruning those picked by $L_1$-norm, confirmed by 86.85% vs. 87.34% for ResNet56 and 65.04% vs. 66.05% for VGG19. As seen, in each run, the regularization-based way also *significantly* surpasses its one-shot counterpart. Although five random runs are still too few given the exploding potential combinations, yet as shown by the accuracy standard deviations, the results are stable and thus qualified to support our finding that the regularization-based pruning schedule is better to the one-shot counterpart.

Table 10: Comparison between pruning schedules: one-shot vs. GReg-1. Pruning ratio is 90% for ResNet56 and 70% for VGG19. In each run, the weights to prune are picked *randomly* before the training starts.

| **ResNet56 + CIFAR10** | Run #1 | Run #2 | Run #3 | Run #4 | Run #5 | Mean$_{\pm\text{std}}$ |
|---|---|---|---|---|---|---|
| Acc. (%, one-shot) | 87.57 | 87.00 | 86.27 | 86.75 | 86.67 | $86.85_{\pm0.43}$ |
| Acc. (%, GReg-1, ours) | **89.26** | **88.98** | **88.78** | **89.42** | **88.96** | $\mathbf{89.08_{\pm0.23}}$ |
| **VGG19 + CIFAR100** | Run #1 | Run #2 | Run #3 | Run #4 | Run #5 | Mean$_{\pm\text{std}}$ |
| Acc. (%, one-shot) | 64.56 | 65.06 | 65.07 | 65.05 | 65.48 | $65.04_{\pm0.29}$ |
| Acc. (%, GReg-1, ours) | **66.63** | **66.57** | **66.80** | **66.80** | **67.16** | $\mathbf{66.79_{\pm0.21}}$ |

