# OpenReview forum: "Neural Pruning via Growing Regularization"
_ICLR.cc/2021/Conference — ICLR 2021 Poster_

### Official Review · AnonReviewer4 · 2020-10-27
**A nice idea for neural pruning with extensive experiments**

**Rating:** 8
**Confidence:** 5

**Review:**

Summary:
The authors propose regularization-based pruning methods with the penalty factors uniformly increased over the training session. The first algorithm (GReg-1) sorts the filters by L1-norm and only applies the increasing regularization to the “unimportant” filters; the second one (GReg-2) applies the increasing regularization to all the filters. The experiments are very extensive and convincing to support the claimed contributions.

Strengths:
1.The idea of utilizing Hessian without knowing its values is interesting to me. Theoretical analysis in Sec. 3.3 looks sound, which should be the main theoretical contribution of this work.

2.Empirical performances are promising, especially the ImageNet one in Tab. 3 and 4.

3.Good performance. On CIFAR-10 and ImageNet, they evaluate the proposed methods with popular deep neural networks, reporting encouraging performances.

4.The methods are easy to implement with current deep learning tools and can work in both the structure pruning and unstructured pruning cases, which is a plus.

5.The authors present theory analyses to show this will cause the filters to separate owing to their different underlying Hessian structure, thus achieving “exploiting the Hessian information without knowing their specific values”.

6.Generally, this paper is well-written with sound proofs for their formulas, although there are some small problems (see the weakness below) that the authors may want to resolve.

Weakness:
1.In Sec. 3.3, it says h11 > h22 leads to r1 > r2, where r is defined by the new magnitude over the old one. But in Fig. 1, the plot is the normalized magnitude stddev. How these two are related to each other is not so straight to me. Could the authors give more explanation about this since it is said in the paper that Fig.1 is an “empirical validation” for the analysis in Sec. 3.3?

2.Some typos and small glitches:
“importance-based one focus more” -> focuses
Appendix: In the last sentence of A.1 “our results (Tab. R3 and R4)” (I didn’t find R3, R4). Please clarify it.
Appendix: Footnote in Tab. 6, the references of [7] [54] seem pointing to nowhere.

===============================After response ====================== The authors have addressed all my concerns.  I would like to keep my initial score.

---

> ### Author Response · Authors · 2020-11-22
> **Author Responses to Reviewer4**
>
> Thanks to R4 for helping improve our paper! We address the concerns as follows.
>
> `R4-Q1`: Weakness: 1.In Sec. 3.3, it says h11 > h22 leads to r1 > r2, where r is defined by the new magnitude over the old one. But in Fig. 1, the plot is the normalized magnitude stddev. How these two are related to each other is not so straight to me. Could the authors give more explanation about this since it is said in the paper that Fig.1 is an “empirical validation” for the analysis in Sec. 3.3?
>
> `R4-A1`:  $r_1 = \frac{\hat{w}_1}{w_1}$, $r_2 = \frac{\hat{w}_2}{w_2}$ (the $*$ mark  indicating the local minimum is omitted here for readability).  $r_1 > r_2$ leads to  $\frac{\hat{w}_1}{w_1} > \frac{\hat{w}_2}{w_2}$ , namely,  $\frac{\hat{w}_1}{\hat{w}_2} > \frac{w_1}{w_2}$. This shows that, after the L2 penalty is increased a little, the new magnitude ratio of weight 1 over weight 2 will be *magnified* if h11 > h22  ($w_1$, $w_2$ are positive in the analysis here, while the conclusion still holds if either of them is negative). As training proceeds, the ratio will grow larger and larger, driving the weights to separate (i.e., the magnitude stddev goes higher), which agrees with what is plotted in Fig.1. Hence, we say Fig.1 is an empirical validation for the analysis in Sec.3.3.
>
> We will add these formulas in the revised version to make the point clear.
>
> `R4-Q2`: Some typos and small glitches: “importance-based one focus more” -> focuses. Appendix: In the last sentence of A.1 “our results (Tab. R3 and R4)” (I didn’t find R3, R4). Please clarify it. Appendix: Footnote in Tab. 6, the references of [7] [54] seem pointing to nowhere.
>
> `R4-A2`: Tab. R3 and R4 simply mean Tab.3 and 4. [7] refers to "Li et al., Pruning filters for efficient convnets, ICLR, 2017". [54] refers to "Liu et al., Rethinking the value of network pruning, ICLR, 2019". We will correct the typos and missed references in the revised version.

---

### Official Review · AnonReviewer2 · 2020-10-27
**Interesting investigation; good performance**

**Rating:** 7
**Confidence:** 5

**Review:**

This paper explores how the basic L2 regularization can be exploited in a growing fashion for better deep network pruning. The authors proposed two algorithms in this work: (1) The first (called GReg-1) is a variant of the L1-norm based filter pruning method [Ref1]. The important/unimportant filters are decided by their L1-norms. Later the unimportant ones are forced to zero through the proposed rising penalty scheme. (2) The second algortihm (called Greg-2) imposes the rising L2 regularization on all the filters. It is theoretically shown in the paper that this makes the parameters to separate to different degrees according to their local curvatures (ie, Hessian values). The method takes advantage of this by driving the weights into two groups with stark magnitude difference and then prunes by the simple L1-norm criterion.  The two methods are demonstrated effective on CIFAR10/100 and ImageNet benchmarks in the comparison with many state-of-the-art methods.

Significance:
The paper focuses on deep neural network pruning, which is of interest to both the academia and industry. The empirical results look significant when compared with related works.

Clarity and quality:
The writing of paper is clear. Important formulas are proved. Related works are properly discussed.

Originality:
The growing regularization method looks novel to me. It further consists of two branches. The first one is simple but the results are inspiring. The second one has a good theoretical basis.

Pros:
The regularization process introduced in GReg-1 allows the network to adapt and recover before some weights are eventually removed. This idea is easy to understand and the investigation of the effect of regularization pruning schedule is insteresting. It is quite surprising that when pruning the same weights, regularization schedule can be quite better than the one-shot manner.

The proposed strategy and its theory basis of GReg-2 seem novel to me. It presents a new way to exploit the Hessian information like no other works before. Although the analyses are two simplified cases, the intuition is straight and also verified empirically on modern deep networks.

The empirical study looks strong on CIFAR10/100 and the large scale ImageNet benchmark.

In addition to structured pruning (filter pruning), they also evaluate their methods in the unstructured pruning case and also achieve good performance compared with the recent pruning paper based on advanced Hessian-approximation.

Cons:
The two methods work pretty well on ImageNet from table 3 and 4, yet one small concern is they are outperformed by C-SGD on CIFAR10 in table 2. Is there more explanation and analysis for this?

Can the proposed methods be generalized to other regularization forms like L1 and L0 [Ref2]? This should be discussed properly in the paper.

Other questions:
How the layer-wise pruning ratios are decided in table 3 and 4 for the proposed methods?

For the Greg-1, it shows pruning the same weights with different schedule schemes can lead to quite different results. One concurrent paper [Ref3] in this venue has a similar finding (different pruning methods with different performances turn out pruning very similar weights). How is your work related with theirs?

[Ref1] Pruning filters for efficient convnets. In ICLR, 2017.
[Ref2] Learning sparse neural networks through l0 regularization. In ICLR, 2018¬¬¬
[Ref3] Rethinking the Pruning Criteria for Convolutional Neural Network, Anonymous, ICLR 2021 submission: https://openreview.net/forum?id=ZD7Ll4pAw7C

In all, I think it’s a good paper and rate as ‘accept’. If most of my concerns and questions are addressed in the rebuttal, I would like to upgrade my rating score.

---

> ### Author Response · Authors · 2020-11-22
> **Author Responses to Reviewer2**
>
> Thanks to R2 for helping improve our paper! We address the concerns as follows.
>
> `R2-Q1`: The two methods work pretty well on ImageNet from table 3 and 4, yet one small concern is they are outperformed by C-SGD on CIFAR10 in table 2. Is there more explanation and analysis for this?
>
> `R2-A1`: A similar concern is brought up by R3 (please refer to `R3-Q4`). Our response there also applies here:
> We conceive this is because CIFAR-10 is a simple dataset and thus the pruned accuracies are already very high (zero accuracy drop by our method and negative accuracy drop by C-SGD), reaching the ceiling. Thus, there is not much difference to show between these methods. The evaluation on the more challenging ImageNet dataset reveals the method difference more faithfully and is also of more practical interest.
>
> `R2-Q2`: Can the proposed methods be generalized to other regularization forms like L1 and L0 [Ref2]? This should be discussed properly in the paper.
>
> `R2-A2`: For GReg-1, it can be easily generalized to other regularization forms like L1. For GReg-2, since the theoretical basis in Sec.3.3 relies on the local quadratic approximation, L2 regularization meets this requirement while L1 does not. Therefore, GReg-2 cannot be (easily) generalized to the L1 regularization as far as we can see currently.
>
> For L0 regularization, it is well-known NP-hard to optimize. In practice, it is typically converted to the L1 regularization case, which we just discussed. We will add these discussions in the revised paper.
>
> `R2-Q3`: Other questions: How the layer-wise pruning ratios are decided in table 3 and 4 for the proposed methods?
>
> `R2-A3`: As discussed in the appendix, we *manually* set these pruning ratios (specific numbers are listed in the appendix) for each experiment to meet a certain speedup target. We do not employ any scheme to automatically decide layer-wise pruning ratio during pruning because the pruning ratio is broadly believed to reflect the redundancy of different layers, which is an *inherent characteristic of the model*, thus should not be coupled with the subsequent pruning algorithms. This choice also makes it easier for following pruning methods to compare with ours when the same layer-wise pruning ratio is used.
>
> Notably, our hand-picking pruning ratios are apparently suboptimal. Yet our methods deliver encouraging performance, still. Better pruning ratios, if found, should advance our methods even more.
>
> `R2-Q4`: For the Greg-1, it shows pruning the same weights with different schedule schemes can lead to quite different results. One concurrent paper [Ref3] in this venue has a similar finding (different pruning methods with different performances turn out pruning very similar weights). How is your work related with theirs?
>
> `R2-A4`: Thanks for pointing out the relevant paper. It presents an interesting finding that using different importance criteria (e.g., L1, L2, GM, Fermat in their paper) the indexes of pruned filters are actually *almost the same*, implying that the performance difference among these methods may not be attributed to a smarter way to find the truly unimportant weights. There is something else at play. Their finding actually partly supports our claim in this paper: Apart from the importance criterion, there are other weight-carrying factors (pruning schedule is one of them as we show) that can lead a big impact on the pruning performance. They may well deserve more research attention, too.

---

### Official Review · AnonReviewer3 · 2020-10-28

**Rating:** 6
**Confidence:** 5

**Review:**

Summary:

The paper deals with regularisation based model compression for DNNs focusing on filter pruning. In particular they focus on growing regularisation during training instead of using a high regularisation parameter at the beginning. Growing regularisation for unimportant weights and negative penalty for important weights help them in improving the expressive power of the important weights.

+ve

- The paper is easy to follow and the authors make it quite clear about the problems related to filter pruning that they are tackling in this work.

- The experiments conducted are quite elaborate and the authors have provided detailed description about the experimental setup, hyper-parameters tuning, and training strategy.

Concerns

- The core idea of growing the regularisation parameter is not quite novel. The paper “Structured pruning for efficient convnets via incremental regularization“ by Wang et. al. also uses different learning rates for different set of parameters as well as they also grow the regularisation during training.

- In table 1, it would good to see a competitive method other than one-shot pruning because it is well established (this paper also mentions it correctly) that pruning a large number of filters in one-shot usually results in significant drop in accuracy. Comparing with iterative pruning methods like “Importance estimation for neural network pruning” by Molchanov et.al. would make the comparison fair especially for higher pruning percentages.

- While comparing GReg1 & GReg2 with other methods in Table 2&3, it would be good to see comparison with other regularisation based methods like “Structured pruning for efficient convnets via incremental regularization“ by Wang et. al.

- In Table 2, the authors should include comparison with Ding et.al. for VGG as well. It seems that for ResNet56 the performance of GReg-1 and GReg-2 is quite comparable to Ding et.al.
- The paper needs thorough proof reading as there are a lot of grammatical mistakes.

- Lastly, the style files used by the authors doesn’t match with the ICLR2021 style files.

---

> ### Author Response · Authors · 2020-11-22
> **Author Responses to Reviewer3 (Part 2)**
>
> `R3-Q3`: While comparing GReg1 & GReg2 with other methods in Table 2&3, it would be good to see comparison with other regularisation based methods like “Structured pruning for efficient convnets via incremental regularization“ by Wang et. al.
>
> `R3-A3`: As suggested, we will add the top-1 accuracy comparison with [*1] on ImageNet, ResNet50 as below. Both of our methods are *significantly* more favorable than [*1] in terms of either absolute top-1 accuracy or accuracy drop.
>
> |  | Speedup | Pruned top-1 acc. | Top-1 acc. drop |
> |-|-|-|-|
> | [*1] | 2.0x | 72.47 | 3.13 |
> | GReg-1 (ours) | 2.31x | 75.16 | 0.97 |
> | GReg-2 (ours) | 2.31x | 75.36 | 0.77 |
> | [*1] | 3.0x | 71.07 | 4.53 |
> | GReg-1 (ours) | 3.06x | 73.75 | 2.38 |
> | GReg-2 (ours) | 3.06x | 73.90 | 2.23 |
>
>
> `R3-Q4`: In Table 2, the authors should include comparison with Ding et.al. for VGG as well. It seems that for ResNet56 the performance of GReg-1 and GReg-2 is quite comparable to Ding et.al.
>
> `R3-A4`: We also noted that on ResNet56 our method does not show an advantage over C-SGD [*3] in our paper (page 6, “our methods perform a little worse than C-SGD...”). We conceive this is because CIFAR-10 is a simple dataset and thus the pruned accuracies are already very high (zero accuracy drop by our method and negative accuracy drop by C-SGD), reaching the ceiling. Thus, there is not much difference to show between these methods.
>
> We did not compare with C-SGD on VGG simply because they do not benchmark this network (their paper does have results on VGG, but the details, e.g., unpruned accuracy and speedup, are missing. Thus, we cannot make a fair comparison). A probably more compelling way to unveil the difference between C-SGD and ours is to benchmark them on a *more challenging* dataset. In this sense, we already have the comparisons on ImageNet, ResNet50 (Tab.3), where our methods are clearly more favorable than C-SGD.
>
> `R3-Q5`: The paper needs thorough proof reading as there are a lot of grammatical mistakes.
>
> `R3-A5`: We will proofread our paper more carefully. It might be more helpful if R3 can *specifically* point out our mistakes.
>
> `R3-Q6`: Lastly, the style files used by the authors don't match with the ICLR2021 style files.
>
> `R3-A6`: We re-checked our template and confirmed that the latest ICLR 2021 latex template has been used. Thus, we do not know how to improve our paper regarding this concern. It might be more helpful if R3 can *specifically* tell us where the style file mismatch problem is presented.
>
> [*1] Wang et al., Structured pruning for efficient convnets via incremental regularization, IJCNN, 2019. (The ResNet50 ImageNet top-1 accuracy results above are obtained from their journal version: https://ieeexplore.ieee.org/abstract/document/8937758)
>
> [*2] Molchanov et al., Importance estimation for neural network pruning, CVPR, 2019.
>
> [*3] Ding et al., Centripetal sgd for pruning very deep convolutional networks with complicated structure, CVPR, 2019.

---

> ### Author Response · Authors · 2020-11-22
> **Author Responses to Reviewer3 (Part 1)**
>
> Thanks to R3 for helping improve our paper! We address the concerns as follows.
>
> `R3-Q1`: The core idea of growing the regularisation parameter is not quite novel. The paper “Structured pruning for efficient convnets via incremental regularization“ by Wang et. al. also uses different learning rates for different set of parameters as well as they also grow the regularisation during training.
>
> `R3-A1`: Although our work shares a general spirit of growing regularization with [*1], the two papers actually are different in many ways:
>
> **Motivation**: The motivations for using the growing regularization are different. [*1] adopts growing regularization to select the unimportant weights by training. Namely, they focus on the *importance criterion* problem. In contrast, we use growing regularization to investigate the *pruning schedule* problem (for GReg-1) or exploit the *underlying Hessian information* (for GReg-2). The importance criterion is simply L1-norm.
>
> **Algorithm design**:
> - [*1] assigns *different* regularization factors to different weight groups based on their relative importance, while we assign them with the *same* factors. For GReg-1, this may not be a substantial difference, while for GReg-2, the difference is fundamental because the theoretical analysis of GReg-2 (Sec.3.3) relies on the fact that regularization factors are kept the same for different weights.
> - [*1] sorts the weight groups *frequently* during training to update the relative importance of weights. This makes their method hard to generalize to the unstructured pruning case because sorting all the weight elements frequently in a DNN demands a lot of time, significantly slowing down the training. In contrast, our methods only do the sorting *once*, with minimal overhead. As a result, our methods can seamlessly generalize to the unstructured pruning (Tab.4) while [*1] cannot.
>
> **Theoretical analysis**: The algorithm in [*1] is generally heuristic-based, while our work provides rigorous theoretical analyses (Sec.3.3) to support the proposed algorithm GReg-2.
>
> **Empirical performance**: Both our methods are significantly better than [*1] on the large-scale ImageNet dataset (please refer to 'Author Responses to Reviewer3 (Part 2) `R3-A3`'  for the specific comparison).
>
> In short, our work is starkly different from [*1] in terms of method motivation, specific algorithm design, theoretical contribution, and empirical performance. These should sufficiently differentiate our work from [*1]. We will add these discussions in the paper to resolve the concern.
>
> `R3-Q2`: In table 1, it would be good to see a competitive method other than one-shot pruning because it is well established (this paper also mentions it correctly) that pruning a large number of filters in one-shot usually results in significant drop in accuracy. Comparing with iterative pruning methods like “Importance estimation for neural network pruning” by Molchanov et.al. would make the comparison fair especially for higher pruning percentages.
>
> `R3-A2`: As stated in the paper (page 5, the last line), Tab.1 is more of an instructive experiment to show the effect of different pruning schedules, than a best-performance benchmark among state-of-the-art methods. Choosing the one-shot scheme to compare is probably the clearest and easiest way to expound our idea *with the irrelevant factors strictly controlled*.
>
> Replacing the one-shot scheme with [*2], as suggested by R3, will make the comparison complex and hard to keep fairness. Two reasons:
>  - [*2] involves stochastic training in determining the weights to prune, i.e., we cannot know which weights will be pruned in advance, while the comparison in Tab.1 needs to know which weights to prune before training.
> - During pruning, there are many irrelevant factors at play that can affect the final results (e.g., the learning rate schedule of [*2] pruning is different from ours). Then we cannot tell if the final accuracy difference is owing to the different pruning schedules or those irrelevant factors. As a result, the comparisons in Tab.1 will totally lose their meaning.
>
> Therefore, we do not compare with [*2] in Tab.1. Instead, we compare with it on the more standard ImageNet benchmark (Tab.3), where our methods are significantly better than it on either ResNet34 or ResNet50, across different speedup ratios.
>
>
> [*1] Wang et al., Structured pruning for efficient convnets via incremental regularization, IJCNN, 2019.
>
> [*2] Molchanov et al., Importance estimation for neural network pruning, CVPR, 2019.

---

### Official Review · AnonReviewer1 · 2020-10-30
**Official Blind Review #1**

**Rating:** 7
**Confidence:** 4

**Review:**

The paper proposes a new pruning scenario using regularization to better prune the network. The scenario has two-component, the first one proposes a new pruning schedule that does not directly remove the neurons that need to prune from the network. It removes the neurons by adding an L2 regularization and makes the neurons that need to remove gradually decrease to zero. The second one gives the importance score to the neurons. It uses the L2 regularization and studies how the coefficient \lambda of the regularization term can influence the weight change to derive the neuron's importance in the neuron network. By perturbing the penalty term to the converged network, the algorithm can get the Hessian information to score the neurons but uses less time than calculating the Hessian. The paper also shows many empirical results on various benchmarks to show their advantages when using the new schedule and scoring criterion during the pruning process. The result shows that their method can get better at a fast speed.

Pros:
I think the paper is interesting, both two steps make sense to me.  For the importance criterion statement, the paper also gives the theoretical analysis. Plus, the paper in good writing and easy to understand.

Cons:
Currently, GReg-1 is based on L1-norm pruning. It would be interesting if the author can show the more empirical result of applying GReg-1 on other pruning methods that directly removes the neurons. It can better support your statement that the pruning schedule is important.


===============================After response ======================
Thanks for stressing my concern, the additional experiment makes the empirical result more convincing. Overall I think this is a good paper, I prefer to keep my score and rate it as accept.

---

> ### Author Response · Authors · 2020-11-22
> **Author Responses to Reviewer1**
>
> Thanks to R1 for helping improve our paper! We address the concerns as follows.
>
> `R1-Q1`: Currently, GReg-1 is based on L1-norm pruning. It would be interesting if the author can show the more empirical result of applying GReg-1 on other pruning methods that directly removes the neurons. It can better support your statement that pruning schedule is important.
>
> `R1-A1`: As suggested, we evaluate the statement that pruning schedule is important with another pruning method OBD [*1], which also removes neurons in the one-shot fashion but adopts an importance criterion derived from Taylor expansion, which is more advanced than the L1-norm.
>
> The results are shown in the table below, where we vary the global pruning ratio from 0.7 to 0.95 so as to cover the major speedup spectrum of interest. Same as Tab.1 in the paper, here the pruned weights are *exactly the same* for the two methods under each speedup ratio. The finetuning process is also the same. Each setting is randomly run 3 times. Mean (std) accuracies are reported.
>
> ResNet56, CIFAR-10:
>
> | Speedup | 2.15x | 3.00x | 4.86x | 5.80x | 6.87x |
> |-|-|-|-|-|-|
> | OBD | 92.90 (0.05) | 91.90 (0.04) | 89.82 (0.11) | 88.56 (0.11) | 86.90 (0.03) |
> | Ours | 92.94 (0.12) | 92.27 (0.14) | 90.37 (0.17) | 89.78 (0.06) | 88.69 (0.06) |
>
> VGG19, CIFAR-100:
>
> | Speedup | 1.92x | 2.96x | 5.89x | 7.69x | 11.75x |
> |-|-|-|-|-|-|
> | OBD | 72.68 (0.08) | 70.42 (0.16) | 62.54 (0.13) | 59.18 (0.32) | 54.19 (0.57) |
> | Ours | 73.08 (0.11) | 71.30 (0.28) | 65.83 (0.13) | 62.87 (0.20) | 59.53 (0.10) |
>
> As seen, using this more advanced importance criterion, our pruning scheme based on growing regularization is still consistently better than the one-shot counterpart. Besides, it is also verified here that a better pruning schedule can bring *more* accuracy gains when the speedup is *larger*.
>
> In addition, in our appendix, we also present the results (Tab.9) of comparing different pruning schemes when a subset of filters is *randomly* picked to prune. GReg-1 is shown to be consistently better than the one-shot scheme in Tab.9, too. This can further support our statement that pruning schedule is important.
>
>
> [*1] LeCun et al., Optimal brain damage, NeurIPS, 1990.

---

### Decision · Program_Chairs · 2021-01-07
**Final Decision**

**Decision:**

Accept (Poster)

**Comment:**

This paper introduces a novel pruning algorithm for neural networks, gently regularizing the weights away (through weight decay) and using Hessian information instead of simple magnitude. All in all an idea that is simple and effective, and could be of interest to a large audience.

AC